# Vibration Reduction of an Overhung Rotor Supported by an Active Magnetic Bearing Using a Decoupling Control System

**Nitisak Numanoy and Jiraphon Srisertpol \***

School of Mechanical Engineering, Suranaree University of Technology, Nakhon Ratchasima 30000, Thailand; numanoy.ni@outlook.com
\*   Correspondence: jiraphon@sut.ac.th; Tel.: +66-4422-4412

**Abstract:** Overhung rotors are important for use in industrial turbo-machines. The effects of a lateral force can increase as a result of the rotor weight, misalignment, or the operating speed of the suspension system for which the rotor is carrying a transmission connection. In this paper, the reduction of vibration in supported lateral directions by varying control is discussed in a radial active magnetic bearing system (AMBS). An experimental test was conducted on the orbital response of an overhung rotor supported by an AMBS, to provide an alternative for improving precision. To simplify the system design, decoupling was achieved using a PID controller and harmonic disturbance compensator (HDC), which improved the rotating performance of an overhung active magnetic bearing (AMB) rotor system, using a frequency response function (FRF) approach and a description of the overhung rotor during normal operational conditions at unique frequencies. The experimental results show that the precision rotation, due to harmonic excitation of the shaft orbit, can be removed in real time using compensation signals using trigonometry. The compensation criteria for the changed run-up and coast-down consistently helped to maintain the rotational center in a central position. A reduction of up to 55% in vibration amplitude on average was achieved under appropriate conditions, and the significance of the overhung rotor symptoms faults were investigated.

**Keywords:** overhung rotor; lateral rotor vibration; AMBS; decoupling control; HDC; FRF approach

## 1. Introduction

In many industrial applications, overhung rotors are used in smart machinery, such as washing machines, helicopter rotors, turbines, generators, centrifuges, pumps, and compressors [1]. Turbomachinery, which is capable of actuating, sensing, and processing information, is widely used. Improvements in turbomachinery, including reliability, utility, and functionality, to enable features such as diagnostics, ultimately leads to improved safety and less required maintenance.

Lateral force caused by unbalanced residual mass is a common issue in rotating machinery [2], but other areas require development, such as misalignment and transmission systems. Unbalance arises if the axis of the rotor is non-coincident with its principal axis of inertia. Perfect balancing in a system is costly, and almost impossible if the distribution of unbalance changes during operation. Thus, residual unbalances always occur. Conventional ball bearings, which can be simplified as stiff spring damper components, constrain the rotor to whirl around its geometric axis. As opposed to conventional bearings, active magnetic bearing systems (AMBSs) may use unbalance compensation. The use of an AMBS as a solution to auto-balance mechanical devices allows the integration of the control into the smart machine during a manufacturing production process. The theory, design, and application of AMBSs have been previously detailed [3]. AMBSs possess several advantages

over mechanical bearings, such as less mechanical wear, low friction, and the absence of pollution by lubrication used to attain high rotational speeds. However, the costs of purchase are several times higher compared with conventional bearings, due to their complexity. Electrical and mechanical engineering and information processing enable the design of AMBSs for a specific application. In larger machines, protection systems, such as back-up bearings, must be mounted on the machine to prevent damage in case of electronic failure or bearing overloading. Therefore, most commercialized rotor systems are equipped with back-up bearings. The stable operation of machines that consist of an AMBS is achieved through the use of suitable magnetic forces generated by a magnetic bearing actuator. The open loop of an AMBS is unstable and requires feedback control for levitation. This added control allows the inertia of the rotor to whirl around its axis, if the space between the stator and rotor in the AMBS is adequately large.

Several authors have investigated active suppression of unbalance vibration in various active magnetic bearing (AMB) rotor systems. Most of the control objectives were to eliminate the synchronous reaction force. The design ideas to compensate for unbalanced forces may be divided into two groups: the gains can be adapted to the stabilization of the loop, which is infinite or high at operating speed [4–6], or a compensation signal can be constructed that removes the harmonic function at sensor measurement [7,8]. As solutions, both structural ideas were fundamentally presented to produce the same result [4]. The rotor was forced to rotate about the principal axis of inertia by identifying the unbalance parameters [9]. However, an accurate model of the control system is needed. Tang et al. [10] used a similar approach with switches to implement the model frequently through the critical speed of the flywheel. The authors, however, did not consider the power amplifier model, which would reduce the complete suppression of unbalance vibration effect at high rotational speeds. This decrease is due to the low-pass characteristic of the power amplifier in the magnetic bearing control system. Changes to and errors in amplifier parameters would directly affect the accuracy of compensation [11]. In addition, the low-pass characteristic is difficult to measure or estimate precisely, because it is nonlinear and varies with time [12,13]. A gain-phase modifier was proposed to achieve adaptive complete suppression of the unbalance vibration by tuning the gain and phase of the feed-forward part, which are adaptively unaffected by the low-pass characteristic of the power amplifier [14,15]. According to the experimental analysis, these algorithms can only be applied after a certain speed. Therefore, detection target faults should be adapted, and higher harmonics have to be determined when the detection targets themselves are in fault. An orbital whirling about a geometric axis is overcomplicated.

This article provides compensation designs and experimental tests of an overhung rotor system for the decoupling control of an orbital whirling about the center of geometry. The assumption for all actuator and displacement sensors is that they are geometrically perfect when installed in the AMBS. In using the principal design idea of a notch, feedback is constructed to compensate for an orbital [16]. Thus, the robustness against uncertain variable speed in the dynamics of the overhung rotor is clarified. In this technique, whirling about the center of mass is more precise, according to whirling the center of the mass. The unbalanced force depends on the speed of whirling along with a variant speed of whirling, which has a more obvious influence than the lateral forces at a unique operational speed. Another characteristic of the design of the PID controller (to create the magnitude of vibration) with a harmonic disturbance compensator (HDC; the relative timing of the synchronous reaction force to motivate) is that the AMBS's actuator and displacement sensor do not have to be concentric circles. In this situation, which is possible if the assembled machinery is imperfectly balanced, the operational speed or the nominal geometry have to change the state. The evaluation of the variation in the shape (pattern), amplitude of the orbital patterns, and an oblique that usually occurs during run-up and coast-down was found to provide a valuable diagnostic.

The remainder of this paper is structured as follows: Section 2 outlines the modelling of an overhung AMB rotor system. Section 3 discusses its extension to the decentralized control of HDC structures and the parameters of the HDC mechanisms. Finally, the experimental setup, orbitals, and results are documented and discussed in Sections 4 and 5.

## 2. Modelling of an Overhung Active Magnetic Bearing Rotor System

The overhung rotor experiment was suspended by a rotor in an AMB rotor system, designed and constructed as a research platform in the System and Control Engineering (SCE) Laboratory at the Suranaree University of Technology (SUT; Nakhon Ratchasima, Thailand). The purpose of this test was to simulate an industrial system and study the control of the rotor's dynamic instability. Modelling the proposed overhung AMB rotor system was achieved by deriving the dynamic equation. An AMB rotor whose rotor and electromagnet are not in direct contact was considered. The control was applied in a decoupled system in the radial direction of an overhung rotor. Reducing the actively controlled degrees of freedom (DOF) can lead to miniaturization of the AMBS. Vibration can be suppressed in such systems by varying control.

An overhung rotor was supported horizontally by the AMBS on one side, while being connected to a power-driven DC motor with a transmission (flexible coupler) on the other, and the hand being held by the ball bearing supported by a whirl backup. The rotor responds to the applied imbalance with a whirling motion. At the center of mass, the rotor is defined as 4-DOF including two translational motions in the radial and rotational directions about its axis. The overhung rotor is controlled by four electromagnets, with the AMBS at one end. In addition, the displacements of the rotor from the equilibrium position are assumed to be very small [17]. The primary structure of a suitable type of AMBS is shown in Figure 1. The forces are contributed by the attractive electromagnet levitation acting on the AMB rotor. This model specifies the *O-xyz* plane fixed in space coordinates. The center of mass of the rotor corresponds to the origin (*O*), and the *z*-axis corresponds to its whirling axis. The linear displacements of the center of mass of the rotor along the *x*- and *y*-axes denote *x* and *y*, respectively; $\theta_x$ and $\theta_y$ are the angular displacements of the whirling axis about the *x*- and *y*-axes, respectively; $x_m$, $y_m$, $x_b$, and $y_b$ represent the displacements of the rotor at the magnetic and ball bearing locations; *n* defines the direction corresponding to the *x*- and *y*-axes; and the electromagnetic force and ball bearing forces are expressed as $f_{mn}$ and $f_{bn}$, respectively, which are the locations of action of each force; and $f_{un}$ is the unbalanced force.

The air gaps at all the four AMBSs are 1 mm. The AMBS's stator and laminated silicon steel sheets are mounted at the end of the rotor shaft for the radical support of the AMBS. The radial thickness of the electrical steel is the same as the width of the pole leg. Steel transformer sheets (Si-steel, Type-50CS1300) with a thickness of 0.5 mm per sheet and a magnetic flux density of 1.64 Tesla, which are used as electromagnets, are recommended. The pole polarization sequence is NS–SN–NS–SN with coils connected, as shown in Figure 2, with two pairs of perpendicular electromagnetic coils installed on the stators (U-shaped). The attractive electromagnetic forces of all coils produce direct current responses in the perpendicular direction. In the symmetrical and uncoupled installed AMBS, all coils had the same turns of pole. Two opposing electromagnets can be operated in 2-DOF for motion directions in differential driving mode.

In practice, an internal reaction acting on the rotor via forces and moments about the mass center are replaced by external forces and moments. Following Newton's second law and the principle of the rotor dynamic behavior:

$$
\begin{aligned}
m\ddot{x} &= f_{mx} + f_{bx} + f_{ux}, \\
I_r\ddot{\theta}_x + \Omega I_a\dot{\theta}_y &= l_b f_{by} - l_m f_{my} - l_u f_{uy}, \\
m\ddot{y} &= f_{my} + f_{by} + f_{uy} - g, \\
I_r\ddot{\theta}_y - \Omega I_a\dot{\theta}_x &= l_m f_{mx} - l_b f_{bx} + l_u f_{ux}.
\end{aligned}
\tag{1}
$$

where the rotational speed $\Omega$ rotates about the whirling *z*-axis; *m* is the rotor mass; $I_r$ and $I_a$ are the transverse and polar mass moment of the inertia of the rotor, respectively; and $l_m$, $l_b$, and $l_u$ represent distances to the AMBS, ball bearing, and unbalanced force from the center of mass, respectively.

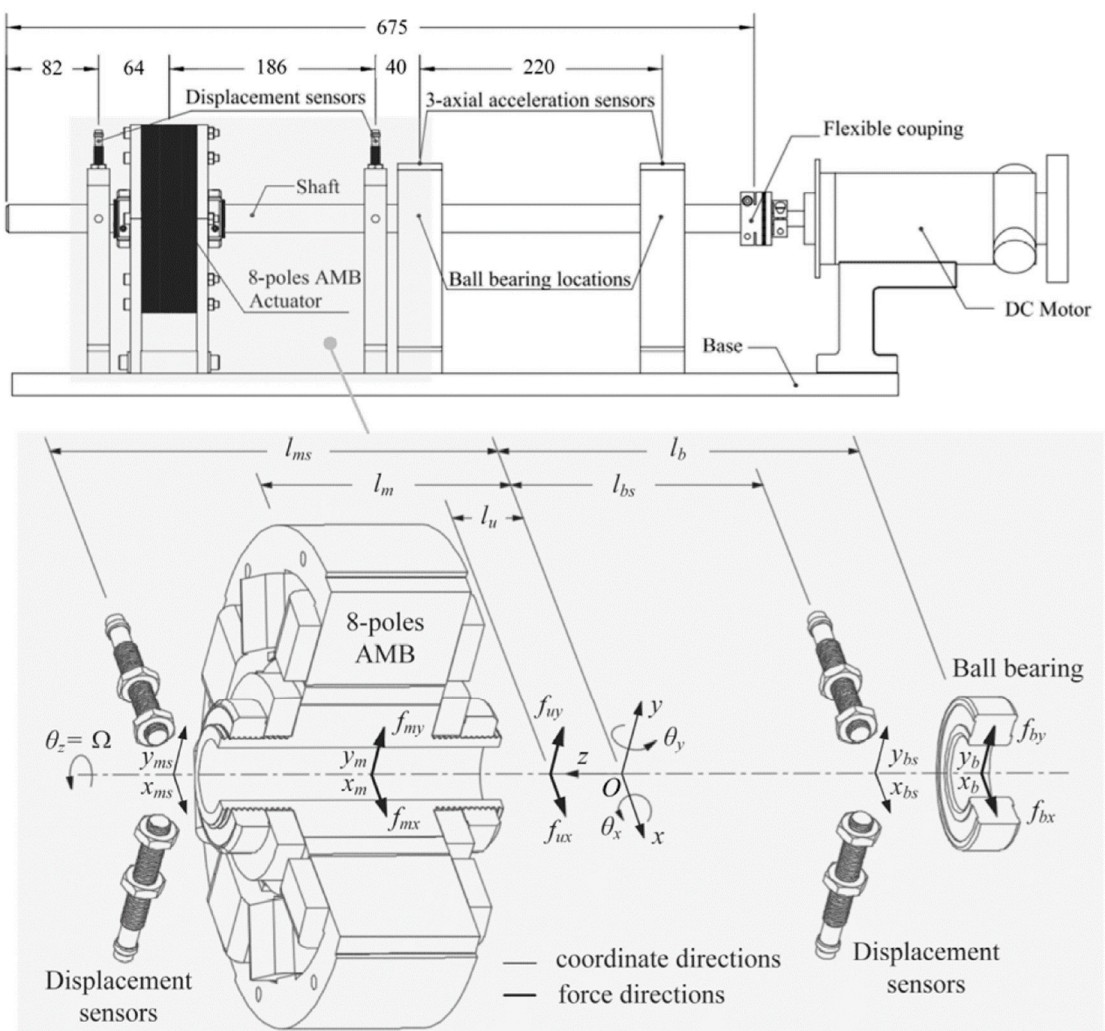

**Figure 1.** Schematic of overhung active magnetic bearing (AMB) rotor.

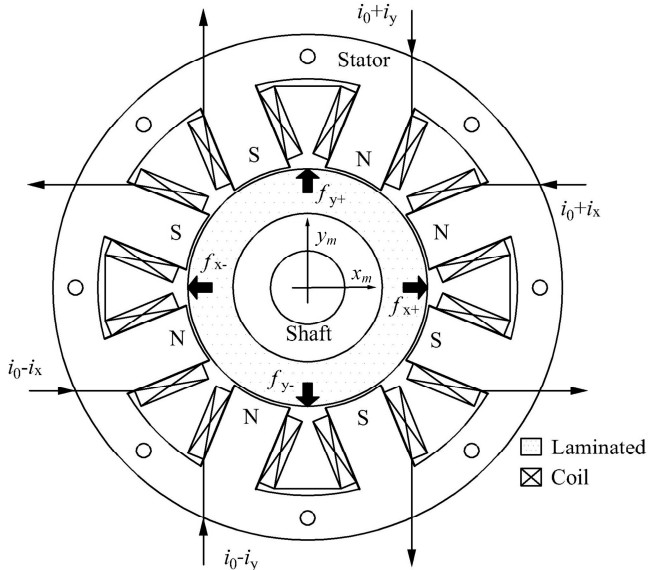

**Figure 2.** Schematic of the pole for driving mode.

To maintain the rotor in the center of the AMBS rotor along the *x*- or *y*-axis, an electromagnetic force was applied to the AMB stator of the overhung rotor. Here, $i_0$ denotes the nominal current, and $i_x$ and $i_y$ are control currents acting on the *x*- and *y*-axes, respectively. Following previous studies [3,11], the total nonlinear forces along the axis of attractive electromagnetism can be determined as:

$$f_{mn} = f_{n+} - f_{n-} = k_m \left( \frac{i_0 + i_n}{n_0 - n_m} \right)^2 - k_m \left( \frac{i_0 - i_n}{n_0 + n_m} \right)^2 = k_s n_m + k_i i_n,$$
$$n \; ; x \text{ or } y,$$
(2)

where $k_m = \mu_0 A N^2/2$, $k_i = 4k_m i_0/n_0^2$, and $k_s = 4k_m i_0^2/n_0^3$ define the current, displacement, and stiffness parameters of the magnets, respectively. We conducted the linearization using a Taylor expansion at the equilibrium point. The coil acting on the *x*- and *y*-axes similarly circulates with nominal current ($i_0$) via Equation (2). Since the nominal air gaps ($n_0 = x_0 = y_0$) are equal along the *x*- and *y*-axes, the displacement and current stiffness parameters that are obtained from the *x*-axis are the same as those obtained for the *y*-axis.

The ball bearing forces neglect the effect of rotation. The stiffness and damper parameters of the ball bearing are expressed as *k* and *c*, respectively, so the bearing supports can be expressed as

$$f_{bn} = -kn_b - c\dot{n}_b$$
(3)

The dynamic equations are in simplest form when Equation (1) expresses the relationship between the magnetic and ball bearing locations in terms of displacements. The assumption is rigid or inflexible for the overhung rotor; the linear and angular displacements from the desired original point are considered very small. We found that the relationship in the plane between stationary coordinates ($x_m$, $x_b$, $y_m$, and $y_b$) and geometry coordinates ($x$, $y$, $\theta_x$, and $\theta_y$) is as shown in Equation (4). Similarly, an additional transformation from the plane of the sensor coordinates ($x_{ms}$, $x_{bs}$, $y_{ms}$, and $y_{bs}$) to the origin *O* can be described as

$$x_m = x + l_m \theta_y, y_m = y - l_m \theta_x, x_b = x - l_b \theta_y, y_b = y + l_b \theta_x,$$
$$x_{ms} = x + l_{ms} \theta_y, y_{ms} = y - l_{ms} \theta_x, x_{bs} = x - l_{bs} \theta_y, y_{bs} = y + l_b \theta_x.$$
(4)

Substituting Equation (2) through Equation (4) into Equation (1) yields the dynamics of the system as follows:

$$\ddot{x}_{ms} - \gamma_1(\dot{y}_{bs} - \dot{y}_{ms}) - k_s(a_1 x_{ms} + a_2 x_{bs}) + k(a_3 x_{ms} + a_4 x_{bs}) + c(a_3 \dot{x}_{ms} + a_4 \dot{x}_{bs}) = b_1 k_i i_x + d_1 f_{ux},$$
$$\ddot{x}_{bs} + \gamma_2(\dot{y}_{bs} - \dot{y}_{ms}) - k_s(a_5 x_{ms} + a_6 x_{bs}) + k(a_7 x_{ms} + a_8 x_{bs}) + c(a_7 \dot{x}_{ms} + a_8 \dot{x}_{bs}) = b_2 k_i i_x + d_2 f_{ux},$$
$$\ddot{y}_{ms} + \gamma_1(\dot{x}_{bs} - \dot{x}_{ms}) - k_s(a_1 y_{ms} + a_2 y_{bs}) + k(a_3 y_{ms} + a_4 y_{bs}) + c(a_3 \dot{y}_{ms} + a_4 \dot{y}_{bs}) = b_1 k_i i_y + d_1 f_{uy} - g,$$
$$\ddot{y}_{bs} - \gamma_2(\dot{x}_{bs} - \dot{x}_{ms}) - k_s(a_5 y_{ms} + a_6 y_{bs}) + k(a_7 y_{ms} + a_8 y_{bs}) + c(a_7 \dot{y}_{ms} + a_8 \dot{y}_{bs}) = b_2 k_i i_y + d_2 f_{uy} - g,$$
(5)

where *s* defines the sensor locations, and

$$l_s = l_{ms} + l_{bs}, \; \gamma_1 = \frac{l_{ms} I_a \Omega}{l_s I_r}, \; \gamma_2 = \frac{l_{bs} I_a \Omega}{l_s I_r}, \; b_1 = \frac{l_{ms} l_m}{I_r} + \frac{1}{m}, \; b_2 = \frac{1}{m} - \frac{l_{bs} l_m}{I_r}, \; d_1 = \frac{l_{ms} l_u}{I_r} + \frac{1}{m}, \; d_2 = \frac{1}{m} - \frac{l_{bs} l_u}{I_r},$$
$$a_1 = \frac{l_{ms}}{I_r}(l_{bs} l_m + l_m^2) + \frac{1}{m}(l_m + l_{bs}), \; a_2 = -\frac{l_{ms}}{I_r}(l_b^2 - l_{ms} l_m) + \frac{1}{m}(l_{ms} - l_m), \; a_3 = \frac{l_{ms}}{I_r}(l_b^2 - l_{bs} l_b) + \frac{1}{m}(l_{bs} - l_b),$$
$$a_4 = -\frac{l_{ms}}{I_r}(l_{ms} l_b + l_b^2) + \frac{1}{m}(l_b + l_{ms}), \; a_5 = -\frac{l_{bs}}{I_r}(l_{bs} l_m + l_m^2) + \frac{1}{m}(l_m + l_{bs}), \; a_6 = \frac{l_{bs}}{I_r}(l_m^2 - l_{ms} l_m) + \frac{1}{m}(l_{ms} - l_m),$$
$$a_7 = -\frac{l_{bs}}{I_r}(l_b^2 - l_{bs} l_b) + \frac{1}{m}(l_{bs} - l_b), \; a_8 = \frac{l_{bs}}{I_r}(l_b l_{ms} + l_b^2) + \frac{1}{m}(l_b + l_{ms}).$$

We carefully considered the assumption of this AMB rotor: $l_s I_r$ is larger than $l_{ms} I_a$ and $l_{bs} I_a$. The two supports in the same direction, creating a gyroscopic effect, and coupling can be ignored. Simultaneously, $|k_s a_1 + k a_3| \gg |k_s a_2 + k a_4|$ and $|c a_3| \gg |c a_4|$ for the magnetic location sensor, and $|k_s a_6 + k a_8| \gg |k_s a_5 + k a_7|$ and $|c a_8| \gg |c a_7|$ for the ball bearing location sensor are satisfied. In the same axis, the coupling between two supports was completely ignored. Therefore, Equation (5) was simplified by

reduction with a decentralized force estimator for the disturbance force. Only part of the dynamic motion correlated with a suspended magnet can be described by

$$\begin{aligned}
\ddot{x}_{ms} - k_s a_1 x_{ms} &= b_1 k_i i_x + u_x, \\
\ddot{y}_{ms} - k_s a_1 y_{ms} &= b_1 k_i i_y + u_y,
\end{aligned} \tag{6}$$

and

$$\begin{aligned}
u_x &= d_1 f_{ux} - k a_3 x_{ms} - c a_3 \dot{x}_{ms}, \\
u_y &= d_1 f_{uy} - k a_3 y_{ms} - c a_3 \dot{y}_{ms} - g, \\
f_{ux} &= m_u e \Omega^2 \cos(\Omega t), \\
f_{uy} &= m_u e \Omega^2 \sin(\Omega t),
\end{aligned} \tag{7}$$

where $u_x$ and $u_y$ are the summation of the disturbance forces, including bearing forces, force of gravity, and unbalanced forces for both the *x*- and *y*-directions, respectively. The transfer function $G_p$ (*s*) is the quotient between the transformed displacement and input current of the decoupling rotor model:

$$G_p(s) = \frac{b_1 k_i}{s^2 - a_1 k_s}. \tag{8}$$

The power amplifier of AMBS can be simplified as an amplifier gain ($g_a$) when connected to a coil:

$$i_n = g_a v_n, \delta \approx \begin{cases} 1 & , & v_n \geq 1 \\ v_n & , & 0 < v_n < 1 \\ 0 & , & v_n \leq 0 \end{cases} \tag{9}$$

The H-bridge (0–$V_s$) is the range of the pulse width modulation (PWM) power signal, where $V_s$ is the supply voltage. The equivalent of the voltage across the H-bridge load is related to the duty-cycle ($\delta$) of its input control voltage $v_n$. Thus, the control voltage is related to the control signal. Conventional controls are available for closing the feedback loop by showing that the coil current $i_n$ is appropriate for the control law [18]. The negative of pole in an open-loop AMBS is unstable when represented by $a_1 k_s$. The most intuitive approach for a conditional control law for an AMB rotor obtained using Equation (8) is implementing a conventional PID control locally for each axis. A general stabilizing controller is designed with a proportion, integrator, and derivative gains as follows:

$$G_c(s) = \frac{v_n}{e_n} = g_p + g_d s + \frac{g_i}{s}, \tag{10}$$

where $g_p$, $g_d$, and $g_i$ are the gains of PID controller.

## 3. Decentralized Control of a Harmonic Disturbance Compensator

The typical unbalance cases for measuring the signal at the sensor input can be modelled as an additional harmonic $u_n$, as shown in Figure 3. The unknowns are the phase and amplitude of $u_n$. The signal of a reference phase and rotation frequency $\Omega$ are assumed to be available. Normally, rotation speed $\Omega$ is constant or varies slowly. This motion is completely dependent on $u_n$, which is an unknown unbalance described by Equation (7). The techniques for unbalance compensation related to the generalized feedback of the notch may narrow the stability margin of the closed-loop system. The negative phase of the notch was used to describe the characteristics of the transfer function that lead to instability below the natural frequency of a rigid mode. Especially for large scales and low speeds, the rotor may experience significant vibrations, caused by an unbalance. Herzog et al. designed a controller to ensure closed-loop stability using the insertion of a notch filter structure. Unlike the conventional structure of the generalized notch filter, we modified the phase-shift angle to substitute the transformation. The stable closed loop of an AMBS can be preserved only by adapting the improvement phase. The internal feedback structure of the HDC's component notch is replaced by

$H(s)$ with a phase shift $\eta_n$, as shown in Figure 3. Let $n_{ms}$ and $c_n$ denote the feedback components of input and output fault signals, respectively. The feedback components of dynamic can be described as

$$c_n = \begin{bmatrix} \sin(\Omega t + \eta_n) & \cos(\Omega t + \eta_n) \end{bmatrix} \int \begin{bmatrix} n_{ms}\sin(\Omega t) \\ n_{ms}\cos(\Omega t) \end{bmatrix} dt. \tag{11}$$

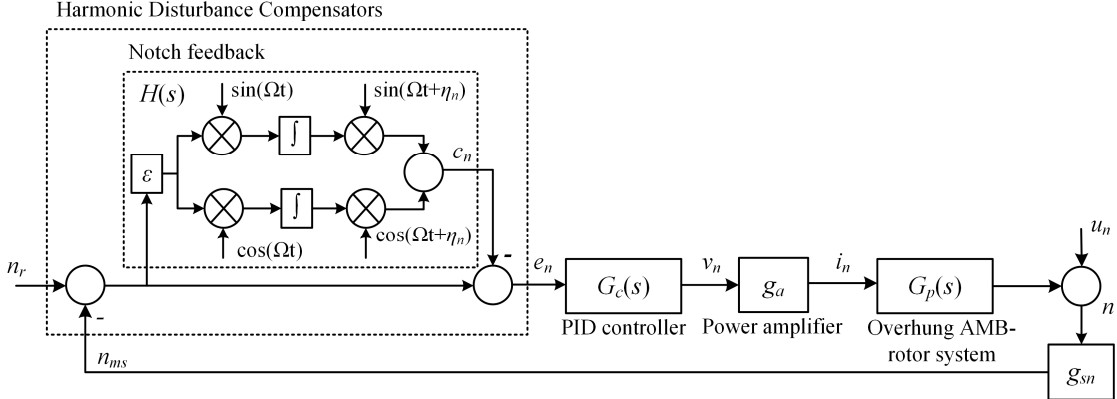

**Figure 3.** Diagram of the HDC connected to a PID controller.

The assumptions $\dot{\Omega} = 0$ and $\dot{\eta}_n = 0$ differentiate Equation (11) with respect to time yields:

$$\ddot{c}_n = -\Omega^2 c_n - \Omega\sin(\eta_n)n_{ms} + \cos(\eta_n)\dot{n}_{ms}. \tag{12}$$

The dynamic equation of the frequency response of $H(s)$ is easy to verify. The Laplace transform is derived as

$$H(s) = \frac{c_n}{n_{ms}} = \frac{\varepsilon(\cos\eta_n - \Omega\sin\eta_n)}{s^2 + \Omega^2}, \tag{13}$$

where $\varepsilon$ is the compensator gain of the HDC. Therefore, the advantages of the notch filter structures with a phase shift can be exploited by the compensator. The design parameters are reduced, so this technique is easier and convenient. The compensator is mainly affected by $\varepsilon$ in the conversion of rotational speed. The phase compensation at the same speed of $\Omega$ can be obtained by selecting proper values determined independently of $\varepsilon$ and $\eta_n$. Figure 3 depicts an unbalance vibration compensation with a PID controller. The transfer function from $n_{ms}$ to $e_n$ is estimated by evaluating Equation (13) at the frequency of $\Omega$, as follows:

$$D(s) = \frac{e_n}{n_{ms}} = (1 + H(s))^{-1} = \frac{s^2 + \Omega^2}{s^2 + \varepsilon\cos\eta_n \cdot s + (\Omega^2 - \Omega\varepsilon\sin\eta_n)} \tag{14}$$

$D(s)$ will disappear if $s = j\,\Omega$. The synchronous controlled current is removed by a signal when the speed is the equivalent of rotational frequency ($\Omega$). The phase of $D(s)$ can be changed by adjusting $\eta_n$, which confirms its notch feedback characteristics. Each direction, without the gyroscopic effect and the coupling between the two-radius AMBS in the same axis, can be separated in two DOF in the $x$- and $y$-axes (radial direction). In the $n$-direction, to simplify the representation, the closed-loop stability with a PID controller was clarified. The other properties behave similarly. The transfer function from $u_n$ to $e_n$ is rearranged as.

$$\frac{e_n}{u_n} = \frac{g_n(s^2 + \Omega^2)S(s)}{(s^2 + \Omega^2) + \varepsilon(\cos\eta_n \cdot s - \Omega\sin\eta_n)S(s)}. \tag{15}$$

Let

$$S(s) = \frac{1}{1 + g_{sn}g_a G_c(s) G_p(s)},\tag{16}$$

where $g_{sn}$ is a displacement sensor gain. Then, the closed-loop stability is determined by roots.

$$(s^2 + \Omega^2) + \varepsilon(\cos\eta_n \cdot s - \Omega\sin\eta_n)S(s) = 0.\tag{17}$$

If $\varepsilon = 0$, then $s = j\Omega$. The differentiable function of $s(\varepsilon)$ at $\varepsilon = 0$ can be expanded with a root locus. For the linearization, starting at $j\Omega$ for $\varepsilon = 0$, the function of root locus yields

$$\frac{\partial s(\varepsilon)}{\partial\varepsilon} = -\frac{1}{2}(j\Omega\cos\eta_n - \Omega\sin\eta_n)S(j\Omega).\tag{18}$$

If the derivatives in Equation (18) are based on the continuity function $s(\varepsilon)$, the stability of the closed loop in the narrow-band case is $\varepsilon \ll \Omega$, and the locations of all poles appear in the left-hand plane (LHP). This enables the use of the continuity in the implicit poles of a closed-loop system as a function of $\varepsilon$. Therefore, all poles of the previous loop $S(s)$ are located in the LHP, and a small $\varepsilon$ exists, so that the root beginning from these poles remains in the LHP. The poles in the critical part of the root occur due to the characteristic equation of $D(s)$. The stability condition in decoupling control for HDC may then be expressed as

$$\frac{\partial s(\varepsilon)}{\partial\varepsilon} = -\frac{1}{2}(j\Omega\cos\eta_n - \Omega\sin\eta_n)S(j\Omega)\tag{19}$$

## 4. Experiment and Results

The proposed control method was applied in experimental tests on an overhung AMB rotor to test effectiveness, as shown in Figure 4. The system consisted of a horizontal rotor, a magnetic bearing, a ball bearing, and a DC motor. The rotor at the ball bearing end was assembled using a flexible coupling to reduce radial forces and torques if the motor and the rotor were not perfectly aligned. The backup ball bearing was placed in the middle point of the rotor between the AMBS and rear ball bearing, which were integrated at the radical support AMBS casings to guard against damage to the AMBS when the rotor drops. We used a thin shaft and one or more heavy AMB rotors to ensure that the eigenfrequencies or natural frequencies of the system were low. This not only reduces the costs of the components, but improves safety. By selecting this setup, two resonance regions of the system of rotor lie in the operating range.

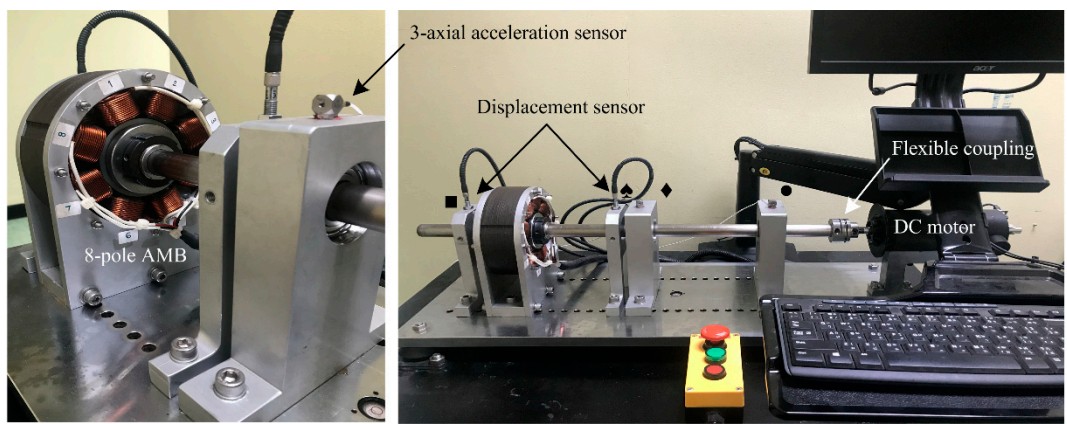

**Figure 4.** Experimental overhung rotor of the AMB rotor system.

The rotating speed in the experiment ran the rotor at 6000 rpm (100 Hz) and 3 hp. For the AMBS actuator, the main parameters and their values are shown in Table 1. As a digital control system to act as a digital signal processor (DSP), we used a DSP board (Maneesoon Group Company Limited, Bangkok,

Thailand). Information for control algorithms were coded in MATLAB, and Simulink software packages and were then codified and uploaded to the DSP board. The sampling frequency was 2 kHz. At the sample time of each channel, four successive 12-bit A/D conversions of each signal were recorded, using four PWM ports and one encoder port. Computer interface circuits were used for command and data communication between the signal and the development module. The corresponding rotor motion was measured via analog inductive sensors to measure the gap (SIEA-M8B-PU-S). The input range of each channel was from 0 to +5 V. H-bridge amplifiers were installed for each coil, and the amplifier featured a 1 kHz switching frequency. The maximum continuous current was rated at 40 A ($V_s$ = 24 VDC). The characteristics of the linearization zone for the relationship between the air gap and the obtainable control current of an AMBS on the $x$- and $y$-axes are shown in Figure 5.

**Table 1.** Parameters of AMBSs, descriptions, and values.

| Description | Parameter | Value (Unit) |
|:---:|:---:|:---:|
| Air gaps | $x_0 = y_0$ | 1 (mm) |
| Pole face area | $A$ | $4.025 \times 10^{-6}$ (m$^2$) |
| Winding number per coil | $N$ | 60 (rev) |
| Bias current | $i_0$ | 2 (A) |
| Range of current to control | $i_n$ | 0–10 (A) |
| Magnetic permeability | $\mu_0$ | $4\pi \times 10^{-7}$ (Vs/Am) |
| Amplifier gain | $g_a$ | 48.3 (A/V) |
| Displacement sensor gain | $g_{ns}$ | 1.18 [I], 1.32 [II] (V/mm) |
| Current stiffness | $k_i$ | 26.09 (N/A) |
| Displacement stiffness | $k_n$ | 198.24 (N/mm) |

[I] $x$-axis, [II] $y$-axis.

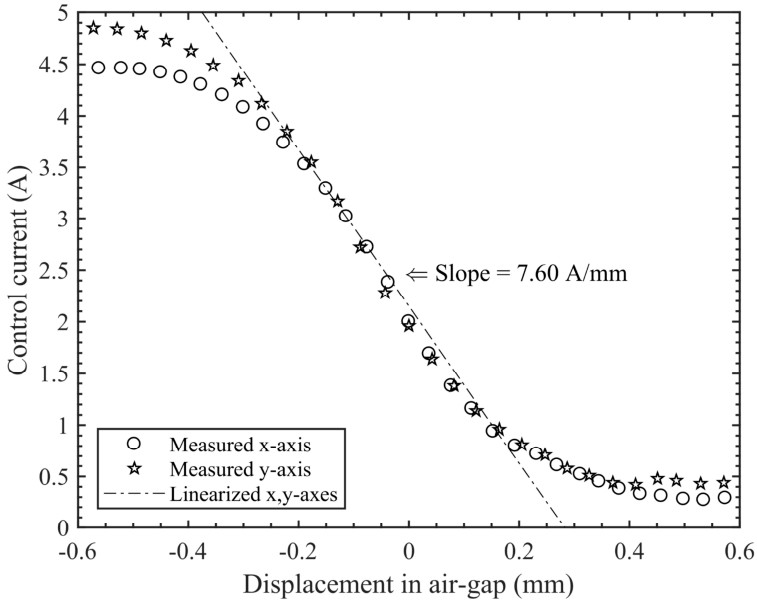

**Figure 5.** Relationship between air gap and obtainable control current of an AMBS on the $x$- and $y$-axes.

### 4.1. Frequency Boundary Detection with Impact Testing

A fundamental measurement of frequency response function (FRF) information was separated from the inherent dynamic properties of the structure. Experimental tests were also conducted from a set of FRF measurements, including frequency, damping, and mode shape. During impact testing, a spectrum analysis is the most popular technique, as it is a convenient, fast, and inexpensive method used to find the modes of structure. The test instrument for performing the operation was a load cell

attached to the head of an impact hammer to measure the input force. An accelerometer was used to measure the magnitude and direction of the response at a fixed point, and a two-channel, fast Fourier transform (FFT) analyzer was used to calculate the FRFs.

The result for the *x*-axis in the identification process with the FRF is shown in Figure 6. We observed clear and separate resonance peaks. In this operation, one three-axial acceleration sensor was placed on the rotor (N7), and an equipment impulse hammer moved through the points (roving hammer technique). This technique is used to determine the impact on the rotor at 13 different axial locations (N1 through N13). The vibration indicator, vibration analyzer, and modal analysis software (the DEWESoftTM FRF module) were used to perform the modal test [19]. The sampling rate was 20 kHz. The signal was adjusted to 8192 lines in the FRF setup. The frequency resolution *Df* was 0.61 Hz. The whole FFT window calculation time was 1.63 s. The modal parameters were extracted, and the corresponding natural frequencies are listed in Table 2. Similarly, the FRF of the *y*-axis matched the frequencies of the *x*-axis unless the amplitudes were different.

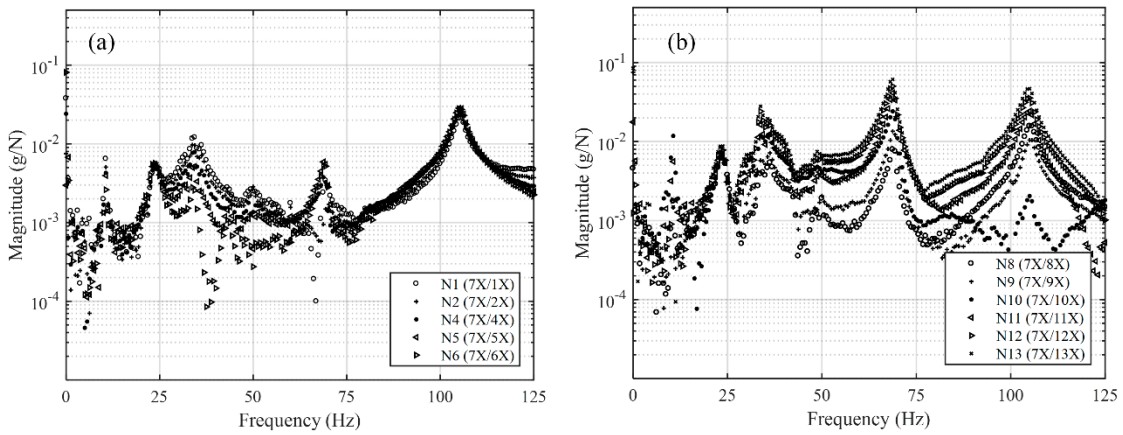

**Figure 6.** An illustration of the FRF associated with the *x*-axis of (**a**) N1 to N6 and (**b**) N8 to N13.

**Table 2.** Experimental frequencies of the frequency response function (FRF) testing.

| FRF Indicator (Hz) | Damping (Hz) | | Critical Damping (%) | |
|---|---|---|---|---|
| | Horizontal | Vertical | Horizontal | Vertical |
| 23.5 | 0.756 | 9.537 | 3.210 | 2.300 |
| 43.6 | 0.445 | n/a | 1.030 | n/a |
| 50.3 | 0.777 | 1.490 | 1.590 | 3.090 |
| 69.0 | 0.845 | 0.988 | 1.230 | 1.430 |

*4.2. Performance Comparison*

We conducted a comparative study of the displacement of convergence rates to examine the performance of the PID, both with and without an HDC controller. The parameters of the PID controller require adjustment for the entire operating speed range, due to the measured displacement signals being impure with other parts (e.g., low-frequency oscillations, disturbances, noises, etc.). However, the different speeds of the AMB rotor depends on the stability of the closed-loop system. Therefore, the closed-loop system may meet the specifications for tracking. Automatic systems modeling using the Simulink software package in MATLAB was applied to define the parameters of the PID controller, using the optimization toolbox. Phase-shift $\eta_n$ was used to choose an appropriate value to satisfy the stability criteria in Equation (19).

We selected optimized values of $\eta_n = 0$ and $\varepsilon = 500$, and a rotor speed lower than 65 Hz under maximum speed, to relate to the peak of FRF in Figure 6. In the response optimization, the maximum overshoots the step response by 5%. Rise time and settling time are less than 0.5 and 1 s, respectively. The controller parameters were optimized to be $g_p = 0.01$, $g_d = 0.0001$, and $g_i = 0.5$. The two sensors were

defined to measure displacement on the *x*- and *y*-axes. The rotor's orbital path was stabilized at the origin point, and was controlled by sustained air to whirl from the PID controller. The representation of rotor vibrations was predominated by rotational speed. The rotational speed of the rotor affected the amplitude. The operations of rotational accuracy depend on the whirling over a range of rotational speeds. The vibrations are created by the unbalanced rotor. The stabilization in the orbital origin can be monitored by the rotor with PID and HDC. The plot contains 10,000 successive samples (in the period at a steady state of 5 s). The improvement in rotation at a constant speed is clear. In the orbit plot, 95% of the operation occurred within the inside of a circular space with a diameter 20 μm, and centered on the orbital origin, as shown in Figure 7.

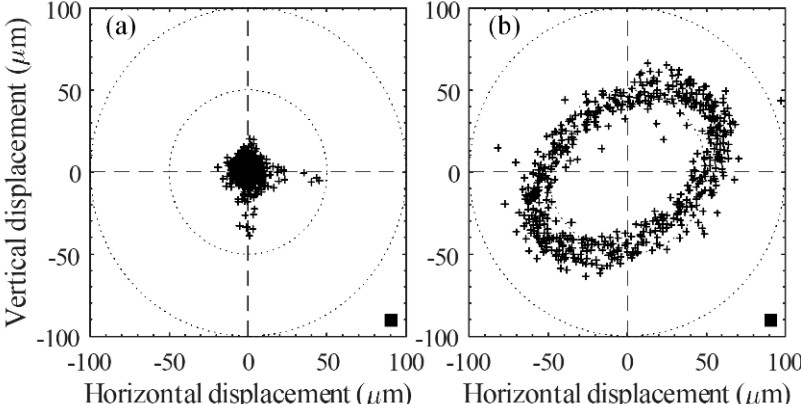

**Figure 7.** Orbit comparisons between (**a**) PID with HDC and (**b**) a PID controller.

Figures 8 and 9 compare the transients of the convergence of displacement amplitudes with the measured values. The values were are extracted from the signals of both axes while the whirling rotor was operating at 35 Hz. The non-convergent time for the PID controller compared with the results provided a convergence time of the PID with the HDC of about eight seconds. This result again demonstrates the superiority of the PID with HDC in terms of convergence rates.

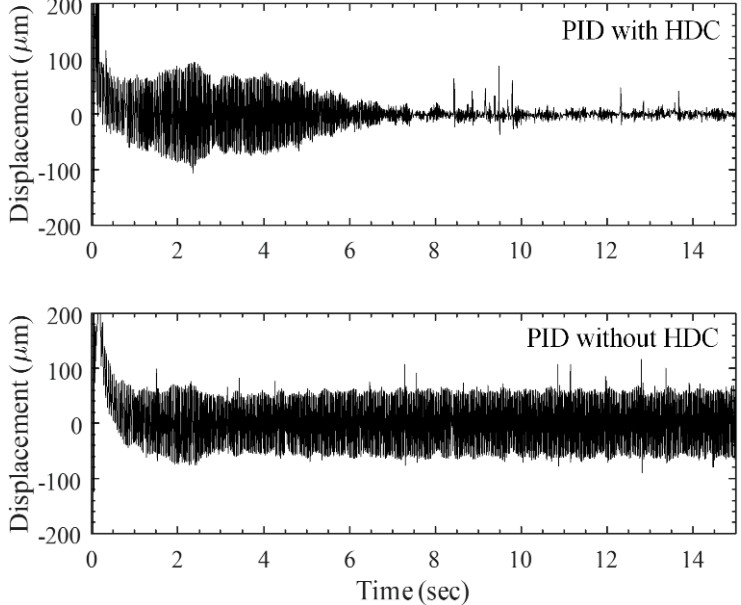

**Figure 8.** Transient of the convergence comparisons on the *x*-axis.

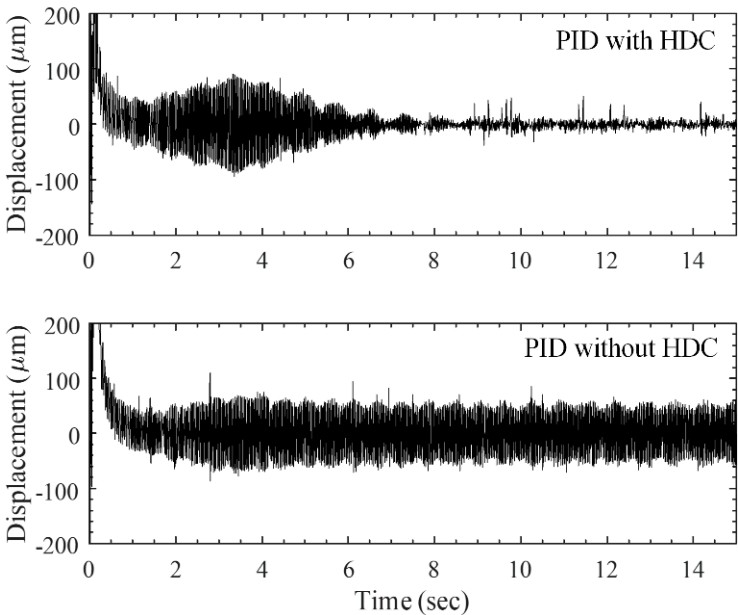

**Figure 9.** Transient of the convergence comparisons on the *y*-axis.

A significant reduction in the vibration amplitude of the shaft was observed, which was compared with the vibration amplitude of both the PID with and without HDC on a magnified scale that was suitable for each rotating speed. The results obtained for the orbit are shown in Figures 10–13 for rotating speeds of 23.5, 43.6, 50.3, and 60 Hz, respectively; each was rotated at a constant subcritical speed of 69 Hz. In these cases, the shaft rotating speed was controlled by a variable frequency drive, and the shaft was subjected to unbalanced excitation. Note that the position of the inductive sensors for non-contact measurements at the AMB rotor and nearby three-axial acceleration sensor points are labeled End-Mag (■) and Mid-Dum (♠), respectively. The Mid-Bear (♦) and End-Bear (●) labels represent the position of the three-axial acceleration sensors that were used for contact measurement in the middle and end of the shafts, respectively.

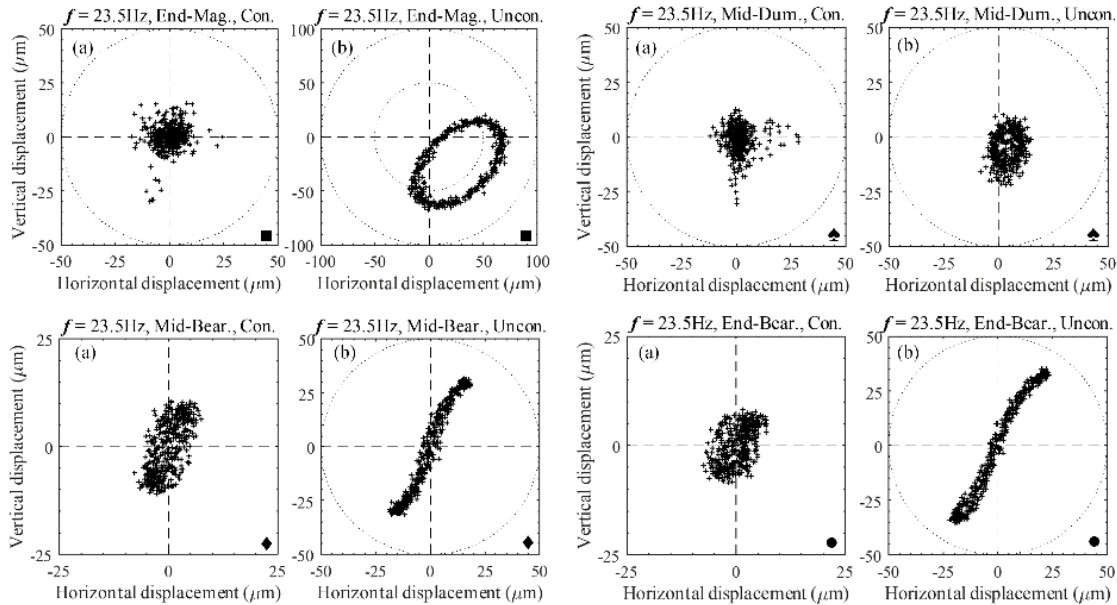

**Figure 10.** Comparison of the orbits measured at a rotating speed of 23.5 Hz, with PID with HDC (**a**) and without HDC (**b**) for each of the sensing points.

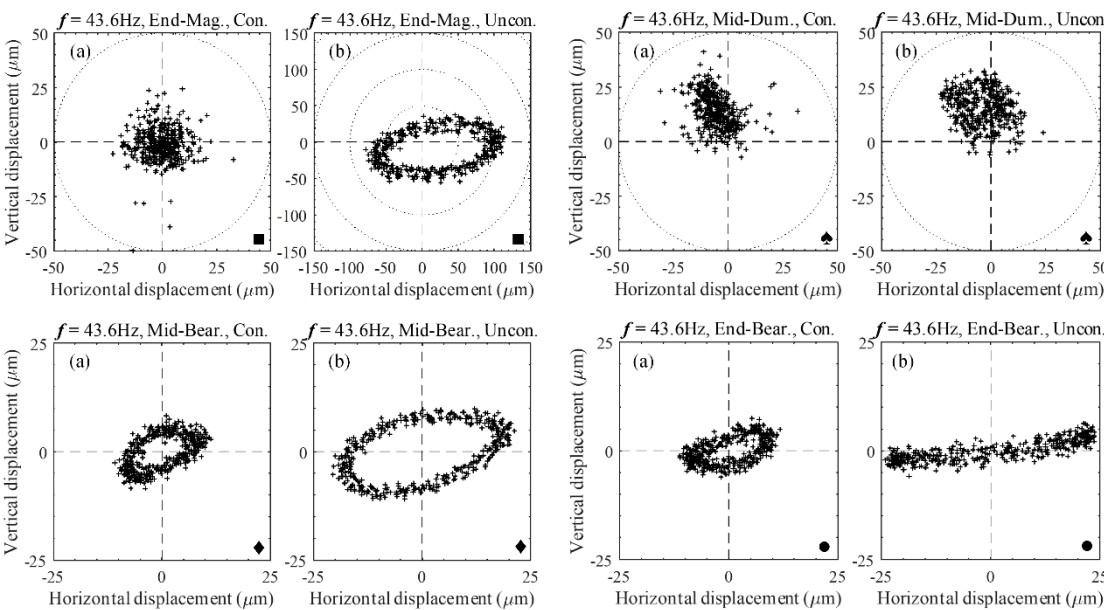

**Figure 11.** Comparison of the orbits measured at a rotating speed of 43.6 Hz, with PID with HDC (**a**) and without HDC (**b**) for each of the sensing points.

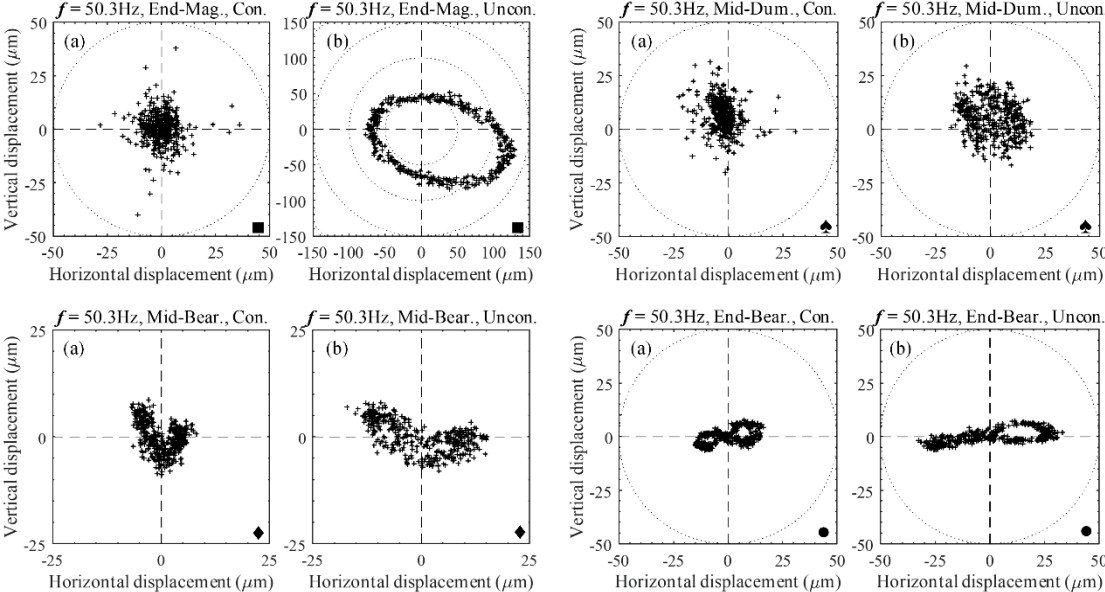

**Figure 12.** Comparison of the orbits measured at a rotating speed of 50.3 Hz, with PID with HDC (**a**) and without HDC (**b**) for each of the sensing points.

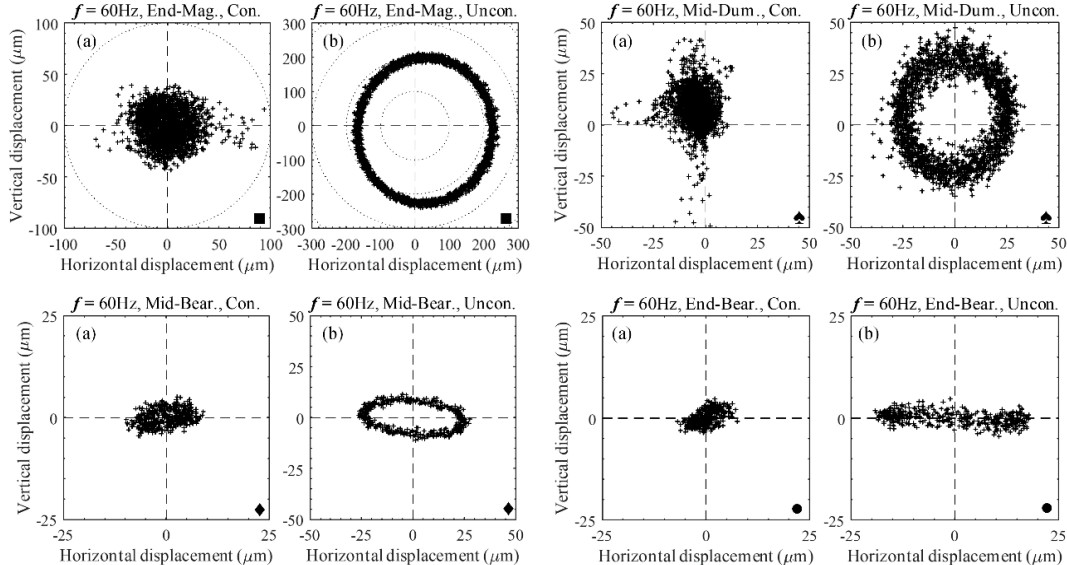

**Figure 13.** Comparison of the orbits measured at a rotating speed of 60 Hz, with PID with HDC (**a**) and without HDC (**b**) for each of the sensing points.

The active system is more effective at all rotating speeds (excitation frequencies). In Table 3, the average values of the shaft's lateral (vertical and horizontal displacements) vibration shows a reduction of about 42–55%. However, as frequency increases, control efficiency decreases, reaching vibration amplitude reductions of about 80% at the AMB rotor point. By observing the orbits of the uncontrolled and controlled shafts, the pure circle of the efficiency of the controller adapts to the frequency increases.

**Table 3.** Percentage of vibration reductions.

| Frequency (Hz) | End-Mag (■) | Mid-Dum (♠) | Mid-Bear (♦) | End-Bear (●) | Average |
|---|---|---|---|---|---|
| 23.5 | 65.4 | 28.6 | 62.5 | 67.2 | 55.9 |
| 43.6 | 80.9 | 36.4 | 32.6 | 33.3 | 45.8 |
| 50.3 | 81.7 | 33.3 | 27.9 | 28.2 | 42.8 |
| 60.0 | 83.2 | 37.5 | 60.2 | 6.6 | 46.9 |

*4.3. Diagnostic of Orbit Shape and Fatigue-Bearing Life Analysis*

In practice, the response of the AMBS is affected by an unbalance or a misalignment; the physical source of these effects may be modelled as a rotor bend and rotor asymmetry [20]. The unbalance vibrations occur at the machine's rotational frequency, but sometimes higher-harmonic rotational speeds are disturbed due to excitation. Machine vibrations caused by an unbalance [21,22] can almost be detected by observing an amplitude and phase shaft displacement, as the machine's rotation whirls through its critical speeds.

The amplitude peaks at the critical speeds and phase changes every 23.5 Hz, passing through about 50.3 Hz at the critical points, which are depicted in the FRF plot in Figure 6. The orbital shaft whirls in an orbit shape or trajectory (unless shaft support impedance exists at the AMB rotor position, in which case, it is circular), and the direction of the ellipse changes when passing through the critical speeds, as shown in Figures 10–13. A side load can be generated at a bearing location as a major indicator of the acting forces. Improper machine assembly can cause a misalignment problem, which affects the supplementary loads on the bearing. The machine's critical speed tends to peak near the vibration problems, unbalance, misalignment, etc. These may either remain unchanged and constant, or increase. In many instances, problems are immediately found when the machine begins to whirl. Then magnitudes increase and disappear equally quickly with an operating speed range.

Intermediate vibration problems of the shaft's amplitude are resolved when the applied magnetic force controls the AMB rotor, so if the orbital rotations were originally circular, the rotations may become elliptical. Because prevention of these issues in the direction opposite the load is easier, the rotor is more inflexible in this direction. Undesirable cases of the orbital rotation transform into banana shapes. (Figure 10, 23.5 Hz, Mid-Bear and End-Bear points) or figure-eight-shaped (Figure 12, 50.3 Hz, End-Bear point). The analysis results show that the loads on the rolling element of the bearing fluctuate significantly under unbalances or the misalignment of different parts of the machine. An increased load from an unbalance results in an inversely exponential reduction in life. Long-standing vibration results in damage and increased breakdown of machines. Affecting their accuracy or performance, these vibrations can also be transmitted to adjacent machinery and machinery subcomponents.

The life of a bearing is closely related to its loads, which are affected by the unbalances and misalignment of its rotating components. The basic life rating of a bearing, according to the international organization for standardization (ISO), is a life expectancy of 90% of the population, where a load life is predicted as 1,000,000 revolutions. It is preferable to calculate life expectancy in operating hours at a constant speed. The basic life equation follows the standard [23]. Typically, life expectancy ranges from months to years at continuous 365-day/24-hour usage. If the load on the bearings is not known, a load ratio of 1:1 is used, for which the life is rated at about 8770 h. If a load ratio is increased to 1:2, the load is increased by 100%. Then, the calculation of life expectancy decreases for ball bearings by 87.7%.

The tuning and improvements of an overhung machine were achieved using the fault identification procedure with this controller. The PID controller with HDC was controlled by the shaft orbit of the rotating machine vibrations in the range of 20 to 60 Hz; the controller design approach for overhung rotors with an AMBS provides a comprehensive high-speed rotor dynamic system. Control of the radical axis is achieved by providing stability and harmonic disturbance rejection, so even high-speed operation can be achieved by the whirling speed and harmonic forcing function, as well as with the feedback from the rotor and sensor locations. The effects of unbalanced compensation for an AMBS with an overhung rotor system, based on the lifespan of the bearings through orbit shape, were studied. The results in Table 3 show that if an increased load considers the average values of vibrations between 42% and 55%, the life of the bearings increases by about 65% to 73%.

## 5. Conclusions

In this study, we applied decoupling to simplify the vertical and horizontal axes of a control loop, in order to reduce the vibration amplitude of the shaft using the electromagnetic force of an AMBS. A PID controller with HDC for an AMB rotor system can maintain the rotational center to approach the center line of an overhung rotor. The analytical and experimental test results demonstrate the effectiveness of the overhung rotor control, and the shaft orbit shape analysis explains the vibration reduction of the overhung rotor in a range of 20 to 60 Hz. Additionally, the test results of the FRF analysis provide the peak frequencies of the predominant magnitude, and we conducted a comparative study between a PID controller with HDC and a conventional PID controller for an AMB-rotor system in terms of shaft orbits, transient responses, and steady-state responses. Abnormalities cause changes to the shaft orbit shape, and the direction of the vibrations precede significant symptoms of faults in an overhung rotor. The main cause of damage is fatigue from substantial vibrations, but the efficiency of the suspension system can be increased using a PID controller and an HDC with AMBS support.

**Author Contributions:** Conceptualization, N.N. and J.S.; methodology and software, N.N.; validation, N.N. and J.S.; formal analysis, N.N.; investigation, J.S.; writing—original draft preparation, N.N.; writing—review and editing, J.S.

**Funding:** This research was funded by Research and Researcher for Industry (RRI) under the Thailand Research Fund (TRF) of Thailand, grant number PHD58I0058.

**Acknowledgments:** The authors are indebted to Suranaree University of Technology (SUT) and the Maneesoon Group Company Limited (MNS), for their generous support and their valuable comments.

**Conflicts of Interest:** The authors declare no conflict of interest.

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
