# Peer review of "Vibration Reduction of an Overhung Rotor Supported by an Active Magnetic Bearing Using a Decoupling Control System"

_machines, doi:10.3390/machines7040073_

Round 1
Reviewer 1 Report
This work focus on vibration control of an overhung rotor and the effectiveness of the active controller has been examined analytically and experimentally. Vibration reduction of this type of rotors is a very important issue especially in rotors with high rotational speed such as microturbines.
The concept of this work is practical, and it is well written. In my opinion, this work has the potential to publish in the journal. The only point that I have is about the formulation. Although the parameters are specified on two figures (Figure 1 and Figure 2), it is just appropriate for geometrical parameters to specify them on figures. For physical parameters in my opinion, it could be more useful if you explained about them after presenting the equations or you could add a nomenclature to the paper.
Author Response
Response to Reviewer 1 Comments
Point 1: The only point that I have is about the formulation. Although the parameters are specified on two figures (Figure 1 and Figure 2), it is just appropriate for geometrical parameters to specify them on figures. For physical parameters in my opinion, it could be more useful if you explained about them after presenting the equations or you could add a nomenclature to the paper.
Response 1: This comment is already completed in an article. We presented the parameters in the paragraph after equations.

Reviewer 2 Report
The paper is interesting from a mechanical practical point of view since it deals about a technology to reduce vibration in turbo-machines. However, there are several lacks and points to improve and clarify:
- The English needs to be considerably improved. There are many typos, the sentences are very long, and the style is unproper. Examples: line 56 "that is are"!?; The paragraph form line 54 to 61 is very badly written. Use punctuation marks properly in all the manuscript.
- In the abstract FRF must be defined.
- The bibliography and background need to be improved. There are few references. What kind of controllers are used in these systems by other authors?
- Between equation (5) and (6) there are symbols that do not look well.
- The work directly provides experimental results. There are no simulations performed with the system model.
- The PID tuning is performed using the optimization toolbox of Matlab. Which model is used for such optimization?
- In this PID tuning optimization, was the HDC included? If so, are the PID parameters of the experiments carried out with the PID without HDC the same that those of PID with HDC? In other words, have you performed different optimizations for the cases: PID with HDC and PID without HDC? This should be more appropriate for performing a fair comparison.
- How is the PID controller implemented?
Author Response
Response to Reviewer 2 Comments
Point 1: The English needs to be considerably improved. There are many typos, the sentences are very long, and the style is unproper. Examples: line 56 "that is are"!?; The paragraph form line 54 to 61 is very badly written. Use punctuation marks properly in all the manuscript.

Response 1: If my manuscript will be available your comments have been replied to edit the English language.
Point 2: In the abstract FRF must be defined.
Response 2: This comment is already completed in an article.
Point 3: The bibliography and background need to be improved. There are few references. What kind of controllers are used in these systems by other authors?
Response 3: Third paragraph of the introduction rewritten to insert the bibliography as
“Several articles and the literatures have investigated active suppression of unbalance vibration in various AMB-rotor systems. Most of the control objectives are merely to eliminate the synchronous reaction force. The design ideas to compensate for the unbalance forces may be divided into two classifications. For the first technique as in [4-6], the gains can be adapted the stabilization of the loop that it is infinite or high at the operating speed. The second is a compensation signal to obviously construct that removes the harmonic function at sensor measurement [7-8]. Although both classifications of structure ideas are fundamentally presented to yield the same thing [4]. The rotor was forced to rotate about the principal axis of inertia by identifying the unbalance parameters [9]. But there is need to be an accurate model of the control system. The technique in [10] used a similar approach with the switches to implement frequently through the critical speed of flywheel. The above method did not consider the power amplifier model which the complete suppression of unbalance vibration effect would reduce at high rotational speeds. This decrease is due to the low-pass characteristic of the power amplifier in the magnetic bearing control system. Meanwhile, the changes and error of amplifier parameters would directly affect the accuracy of compensation [11]. In addition, the low-pass characteristic is difficult to measure or estimate precisely because it is nonlinear and time-varying [12-13]. A gain phase modifier was proposed to achieve adaptive complete suppression of the unbalance vibration by tuning the gain and phase of the feed-forward part, which are adaptively unaffected by the low-pass characteristic of power amplifier [14-15]. According to the experimental analysis, these algorithms can only be achieved after a certain speed. Since detection target faults should be adapted and higher harmonics have to be decided, when the detection targets themselves are in fault. An orbital whirling about a geometric axis are more overcomplicated. ” .
Point 4: Between equation (5) and (6) there are symbols that do not look well.
Response 4: This comment is already completed in an article.
Point 5: The work directly provides experimental results. There are no simulations performed with the system model.
Response 5: In this work, the decoupling to simplify of a control loop form its vertical and horizontal axes can refer to using a simpler model when the correct model is difficult to use. An approximate model is used to make results easier in the actual experimental tests. Approximations might also be used if incomplete information prevents use of exact representations.
Point 6: The PID tuning is performed using the optimization toolbox of Matlab. Which model is used for such optimization?
Response 6: For an active magnetic bearing with PID tuning an external static load will always result in a change of the steady position or equilibrium point. The error signal, i.e. the difference between the reference command input signal and the measured position signal (displacement sensor), is fed into the controller. By expressing the control current (to force coils) by the error signal and by simultaneously applying the control law in optimization toolbox, the following dependency can be obtained for the static load. In the response optimization, the conditions used an overshoot in the step response, rise time and settling time of fixed load to force the rotor, respectively.
Point 7: In this PID tuning optimization, was the HDC included? If so, are the PID parameters of the experiments carried out with the PID without HDC the same that those of PID with HDC? In other words, have you performed different optimizations for the cases: PID with HDC and PID without HDC? This should be more appropriate for performing a fair comparison.
How is the PID controller implemented?
Response 7 In this article does represent to compare between both controllers in terms of implements. The PID control is ubiquitous. The HDC control involves responsibility that implement the PID controller on a target of changing load.
The parameters “P” and “D” of the control law are determined by choosing appropriate values for stiffness and damping of the closed-loop system. Along with the maximum force (load capacity) of a magnetic bearing the bearing stiffness is one of the most basic bearing parameters and should already be defined in the early stages of a magnetic bearing, since the design of important system components such as the bearing size and the amplifier power rating depend on this selection. Evidently, the choice of the closed-loop stiffness underlies the specifications of a particular application. An integrating feedback “I” can be implemented. In the steady state, all signals within the control loop are constant, hence, the error signal must be identically zero. The position measurement signal exactly follows the position reference command input signal, independently of the external load as long as this load is constant. It is important to keep in mind, however, that this is only true for the steady state: dynamically, the error signal will not be zero but will depend on the various time constants in the loop.
The term Harmonic Disturbance Compensators (HDC) reflect the fact that the control output of these schemes has to be implemented to the unknown rotor unbalance, dependently of the external load as long as this load is variation at unique operating speed. This also makes clear why any unbalance control scheme must provide specific constraints of PID with HDC to its system parameters in order to achieve closed-loop stability of the process.
The output of the controller is considered a command signal for the power amplifier of PID controller device and HDC (phase trigger), which has to transform this signal into the physical current flowing through the electromagnet’s coil. In this case the power amplifier is configured as a current amplifier, a control scheme which is the most widely implemented in industrial active magnetic bearing systems. It is easy to implement in most types of industry.

Round 2
Reviewer 2 Report
The English still needs to be considerably improved.
Author Response
We has undergone English language editing by MDPI.
